# Time-reversal symmetry breaking hidden order in $Sr_2(Ir,Rh)O_4$

Jaehong Jeong[1], Yvan Sidis[1], Alex Louat[2], Véronique Brouet[2] & Philippe Bourges[1]

Layered $5d$ transition iridium oxides, $Sr_2(Ir,Rh)O_4$, are described as unconventional Mott insulators with strong spin-orbit coupling. The undoped compound, $Sr_2IrO_4$, is a nearly ideal two-dimensional pseudospin-1/2 Heisenberg antiferromagnet, similarly to the insulating parent compound of high-temperature superconducting copper oxides. Using polarized neutron diffraction, we here report a hidden magnetic order in pure and doped $Sr_2(Ir,Rh)O_4$, distinct from the usual antiferromagnetic pseudospin ordering. We find that time-reversal symmetry is broken while the lattice translation invariance is preserved in the hidden order phase. The onset temperature matches that of the odd-parity hidden order recently highlighted using optical second-harmonic generation experiments. The novel magnetic order and broken symmetries can be explained by the loop-current model, previously predicted for the copper oxide superconductors.

[1] Laboratoire Léon Brillouin, CEA-CNRS, Université Paris-Saclay, CEA Saclay, Gif-sur-Yvette 91191, France. [2] Laboratoire de Physique des Solides, Université Paris-Sud, Université Paris-Saclay, Orsay 91405, France. Correspondence and requests for materials should be addressed to J.J. (email: jaehong.jeong@cea.fr) or to P.B. (email: philippe.bourges@cea.fr).

In the $5d$ layered perovskite material, $Sr_2IrO_4$, spin-orbit coupling and strong electron correlations combine to give rise to a spin-orbit coupled Mott insulator with a pseudospin $J = 1/2$ antiferromagnetic (AFM) state[1,2]. It exhibits close structural[3,4], electronic[5,6] and magnetic[3,4] similarities with the $3d$ layered perovskite material, $La_2CuO_4$, which evolves from a spin $S = 1/2$ AFM Mott insulator to a high temperature superconductor upon doping. Doped $Sr_2IrO_4$ has then become a quite promising material to discover new states of matter, including unconventional superconductivity.

The crystal structure of the layered perovskite $Sr_2IrO_4$ is characterized by the stacking of $IrO_2$ and $SrO$ layers. The $Ir^{4+}$ ion is at the centre of an oxygen octahedron, rotated by $\theta = 11°$ in the basal $ab$ plane. (Fig. 1c). For a long time, $Sr_2IrO_4$ has been described as possessing a tetragonal centrosymmetric structure with four-fold rotational symmetry about the $c$-axis, corresponding to a space group $I4_1/acd$[7]. However, a structural distortion exists even above room temperature as shown by optical second-harmonic generation (SHG) studies[8] yielding to the space group $I4_1/a$ where the $c$- and $d$-glide planes are lost. That gives rise to additional weak Bragg spots such as $(1, 0, 2n + 1)$ that have been reported using neutron diffraction[3,4]. In this material, crystal field effects, spin-orbit coupling, Coulomb repulsion and the bending of the Ir-O-Ir bonds play an important role to understand electronic and magnetic properties.

Within the octahedral crystal field, the Ir $5d$ electronic levels split into $e_g$ and $t_{2g}$ states. Under strong spin-orbit coupling, the $t_{2g}$ states split into a $J = 1/2$ doublet and $J = 3/2$ quartet, so that, among the five $5d$ electrons of the $Ir^{4+}$ ion, only one remains in the $J = 1/2$ doublet state. A large enough effective Coulomb repulsion finally localizes the $J = 1/2$ electron and one is left with a $J = 1/2$ pseudospin model. Below $T_N \sim 230\ K$ (ref. 1), an AFM order develops, characterized by a magnetic propagation wavevector $\mathbf{q}_m = (1, 1, 1)$ and magnetic moments at the Ir sites aligned in the basal $ab$-plane[3,4,9] (Fig. 1c). The directions of the staggered magnetic moments are tied to $IrO_6$ octaedra and follow their rotation, $\theta$, giving rise to a canting of the AFM structure in each $IrO_2$ layer with a ferromagnetic component along the $b$-axis. As shown in Fig. 1c, the magnetic structure of $Sr_2IrO_4$ has a staggered stacking along the $c$-axis of the canted AFM layers

(defined as $+$ or $-$ depending of the sign $\pm\theta$ of the tilt). In other words, the weak ferromagnetic component in each layer is stacked to give a $- + + -$ structure along $c$-axis (AF-I phase)[3,4], so it cancels out globally. However, the AFM interaction along the $c$-axis is rather weak, thus it is easily transformed to the $+ + + +$ stacking order (AF-II phase) by applying an external magnetic field[2]. At variance with the AF-I order, the AF-II stacking is then characterized by a magnetic propagation vector $\mathbf{q}_m = 0$ (ref. 10) and a net (but weak) ferromagnetic moment. Throughout the paper, we refer to these AFM phases, as the one depicted in Fig. 1c, as conventional AFM phases.

Interestingly, the AFM transition temperature is suppressed under the substitution of Rh for Ir. While the Rh substitution could be thought to be iso-electronic, it should be stressed that Rh is likely to play the role of an acceptor ($Rh^{3+}/Ir^{5+}$) and effectively give rise to a hole-doping of the material[10]. Figure 1a shows the magnetic phase diagram in $Sr_2Ir_{1-x}Rh_xO_4$. From bulk magnetization[11-13] and neutron diffraction measurements[3,14], the conventional AFM transition decays almost linearly as a function of Rh substitution up to $x_c \approx 0.15$. Actually, X-ray studies[10] indicate a slightly larger critical doping $x_c \approx 0.17$ with the occurence of another transition at lower temperature when the magnetic correlation lengths are effectively diverging. Further, upon Rh substitution, the AFM order undergoes a transition from the AF-I to the AF-II phase, even at zero magnetic field[10,14].

Recently, a hidden broken symmetry phase, developing prior to the AFM state, has been reported in $Sr_2(Ir,Rh)O_4$ using rotational anisotropy optical SHG measurements[15]. The hidden broken symmetry phase was observed distinctively at a few K above $T_N$ for the pure sample and far above for the doped systems[15]. These data highlight an odd-parity hidden order as both the inversion and four-fold rotational symmetries are broken below a temperature $T_\Omega$ distinct from the Néel temperature $T_N$ (Fig. 1a)[15]. From the symmetry analysis, the SHG results could be in principle explained by a triclinic distortion of the crystal structure. However, there is no experimental evidence using X-ray and neutron scattering of any structural distortion[2-4]. It should be nevertheless stressed that these diffraction studies use too modest spatial resolution to definitively prove a lack of symmetry lowering.

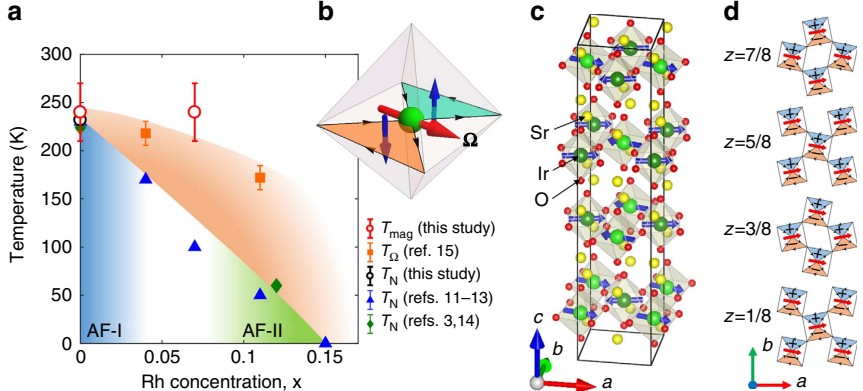

**Figure 1 | Magnetic phase diagram and LC order.** (**a**) Magnetic phase diagram of $Sr_2(Ir,Rh)O_4$. The obtained transition temperatures, $T_N$ and $T_{mag}$ in this study are represented by black and red empty circles for the conventional AFM and the hidden magnetism preserving the lattice translation, respectively. The hidden broken symmetry phase, $T_\Omega$ observed by SHG[15] (orange squares) is also represented as well as the AFM phase determined by magnetization measurements[11-13] (blue triangles) and by neutron diffraction[3,14] (green diamonds). (**b**) A schematic picture for co-planar LC state in a single $IrO_6$ octahedron. The blue and red arrows denote magnetic moments generated by circulating currents and the anapole, $\Omega$, respectively. (**c**) Atomic and AFM structures of $Sr_2IrO_4$ with a space group $I4_1/a$ (origin choice 2) (Ir atoms are represented in green, Sr in yellow and oxygen in red). (**d**) The co-planar LC ordered state in the basal plane. The red arrows denote the anapole vector, $\Omega$, as shown in **b**, and the plus/minus signs correspond to the orbital magnetic moments perpendicular to the $ab$-plane. This nearly-ferro-toroidal order preserves translational symmetry of lattice but breaks parity inversion, four-fold rotational and time-reversal symmetries.

Alternatively, the SHG signal could be due to a magnetic ordering although there is no proof of time-reversal symmetry breaking at $T_\Omega$. Actually, the AF-I ground state of pure $Sr_2IrO_4$ preserves the parity inversion symmetry and thus cannot explain the SHG signal. Instead, a few magnetic point groups that preserve the translation symmetry of the lattice were proposed to account for the SHG signal, such as $2'/m$ or $m1'$ (refs 15,16). In particular, a so far non-observed AFM state of $2'/m$ symmetry, corresponding to a stacking $+-+-$ along the $c$-axis of AFM planes, would produce the SHG signal[16].

Among the magnetic point groups, it is argued[15] that the new broken symmetries can be caused by a loop-current (LC) phase[17,18], theoretically proposed to account for the pseudogap physics of superconducting cuprates. The existence of such a magneto-electric state has gained support from the detection in several cuprate families of its magnetic fingerprint by polarized neutron diffraction[19–24]. Using the same technique for $Sr_2(Ir,Rh)O_4$, we here report at the temperature of the odd-parity order, $T_\Omega$, the appearance of a hidden magnetic order, which breaks time-reversal symmetry while preserving lattice translation invariance. Among the magnetic models inferred from the SHG data[15], only the co-planar LC order[17,18] produces a magnetic diffraction pattern consistent with our polarized neutron data. Our results show that exotic magnetic orders with the same symmetry properties as the LC phase exist in both iridates and cuprates.

## Results

**Momentum location of LC and AFM orders.** Let us first describe the co-planar LC order. It is characterized by two circulating currents turning clockwise and anticlockwise within the same plane inside the $IrO_6$ octahedron (Fig. 1b) and belongs to a $2'/m$ point group symmetry. It breaks time-reversal and inversion symmetries but not their product. The LCs produce two opposite orbital magnetic moments within each $IrO_6$ octahedron. A toroidal pseudovector, or anapole, is defined as an order parameter by $\Omega = \sum_i \mathbf{r}_i \times \mathbf{m}_i$, where $\mathbf{r}_i$ and $\mathbf{m}_i$ stand for the position and the orbital magnetic moment, respectively, in one octahedron (Fig. 1b). It is similar to the toroidal moment in multiferroic systems[25]. When the anapole vectors are stacking parallel along the $c$-axis (ferro-toroidal coupling) as shown in Fig. 1d, the ordered structure does not break translational symmetry but breaks parity inversion and four-fold rotational symmetries[15]. Since the direction of the anapole is bound to the orientation of each $IrO_6$ octahedron, the resulting order is a nearly-ferro-toroidal order (that is, weakly distorted) (Fig. 1d).

As explained in Supplementary Note 1, the Fourier transform of the magnetic correlation function associated with the LC phase is located in iridates at nuclear Bragg peaks, such as $(1, 1, 2+4n)$. These Bragg peaks respect both the original body-centred tetragonal structure condition $H+K+L=2n$ and the $2H+L=4m$ condition due to the $(1/2, 0, 1/4)$ translation. (The wave-vector is denoted by $\mathbf{Q}=(H, K, L)$, see Methods section). This produces a very specific magnetic diffraction pattern, which can be probed by magnetic sensitive diffraction technique, such as neutron scattering technique.

We have investigated both the conventional AFM orders and the LC magnetic order using neutron scattering diffraction experiments. Note that the momentum transfers $\mathbf{Q}$ are different for both types of phase: $(1, 0, L)$ for AFM peaks and $(1, 1, L)$ for the LC peaks (see Supplementary Note 1 and Supplementary Fig. 1). Three major wave-vectors positions, $\mathbf{Q}$, have been examined: their projection onto the reciprocal $HK$-plane are shown in Fig. 2a. We have then studied each phase in a different scattering plane: $H00$-$00L$ for conventional AFM orders and

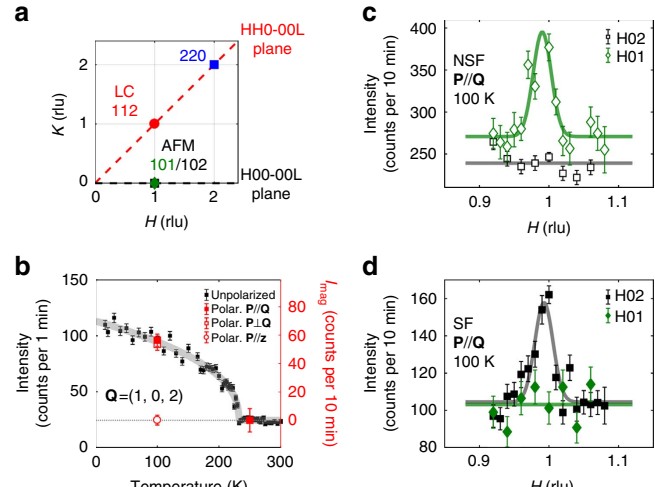

**Figure 2 | AFM order in $Sr_2IrO_4$.** (**a**) Momentum transfer $\mathbf{Q}$ positions (projected in the reciprocal $HK$-plane) investigated with neutron scattering: $(1, 0, L=1, 2)$ corresponds to the AFM Bragg peaks, whereas $(1, 1, 2)$ is where a magnetic scattering is expected for the LC phase. (**b**) Thermal evolution of the peak intensity at the AFM Bragg peak $(1, 0, 2)$ measured by unpolarized neutron (black) and the full polarization analysis at $T=100$ and $250\,K$ (red) (see text). (**c,d**) The $(H, 0, L=1, 2)$ scans at $100\,K$ for both the non-spin-flip (NSF) (empty symbols) and spin-flip (SF) (full symbols) channels (diamond symbols for $L=1$ and square symbols for $L=2$). While the $(1, 0, 2)$ peak (black squares) shows purely magnetic behaviour, the $(1, 0, 1)$ (green diamonds) shows a purely nuclear signal.

$HH0$-$00L$ to study the LC order (see Methods section). We have investigated two iridate samples: a pure $Sr_2IrO_4$ and a 7% Rh-doped one. Small single crystals have been co-aligned to get a large enough sample mass in order to perform the polarized neutron experiment. The sample preparation and magnetization measurements, which characterize the samples, are described in the Methods section (see also Supplementary Fig. 2).

**Conventional AFM orders.** We have first studied the conventional AF-I order by repeating unpolarized neutron diffraction measurements[3,4]. We report in Fig. 2b the AFM Bragg peak at $\mathbf{Q}=(1, 0, 2)$ in the pure system where the magnetic Bragg intensity shows a sharp AFM transition at $T_N=232\,K$. The fitted critical exponent $2\beta\approx0.41$ is consistent to 0.36 in the previous report[3]. Next, we have performed polarized neutron experiment to study the $\mathbf{Q}=(1, 0, L)$ Bragg peaks to disentangle the magnetic and nuclear contributions of these peaks. The polarized neutron experiment setup is presented in the Methods section. It has been already used in previous measurements in cuprates and described in refs 19–24. For a given neutron polarization $\mathbf{P}$, the neutron intensities, $I_{SF}$ and $I_{NSF}$, are measured in both spin-flip (SF) and non-spin-flip (NSF) channels, respectively. Full magnetic signal appears in the SF channel when $\mathbf{P}\|\mathbf{Q}$ (refs 22,24) whereas the nuclear intensity occurs in the NSF channel. Figure 2c,d shows the wave-vector scans along $H$ across the magnetic peak $\mathbf{Q}=(1, 0, 2)$ for the NSF and SF channels. That proves the magnetic origin of the peak as it is only seen in the SF channel (Fig. 2d). In Fig. 2b, the magnetic intensity $I_{mag}=I_{SF}-BG$ is determined at two temperatures, 100 and $250\,K$, below and above $T_N$ (where BG stands for the flat background of Fig. 2d). A clear magnetic intensity is sizeable at $100\,K$, whereas no magnetic intensity is seen at $250\,K$. Further, we report at $100\,K$ in Fig. 2b the magnetic intensity for three different neutron polarization states: the polarizations $\mathbf{P}\|\mathbf{Q}$ and $P\perp\mathbf{Q}$ are in the scattering plane (see Methods section), whereas $\mathbf{P}\|\mathbf{z}$ is perpendicular to the

scattering plane. In the given geometry, zero magnetic intensity with $\mathbf{P}\|\mathbf{z}$ proves that the AFM moments are confined in the $ab$-plane. This is in agreement with the previous studies of the neutron structure factors[3,4].

In the same scattering plane, the Bragg peaks at $\mathbf{Q}=(1, 0, L)$ for $L=1$ and 3 were also measured using polarized neutrons. They correspond to the forbidden peaks of the original $I4_1/acd$ structure giving rise to the space group $I4_1/a$ where the glide planes are lost[8]. As it has been already discussed[3,4], their temperature dependence exhibits no anomaly neither at the Néel temperature nor at the onset of the odd-parity hidden order ($T_\Omega$)[15]. The origin of that scattering is non-magnetic, as shown in Fig. 2c,d. For the Bragg peak at $\mathbf{Q}=(1, 0, 1)$, no signal is sizeable in the SF channel and only a NSF signal is observed above the background. These results exclude the possibility of an AFM arrangement of $+-+-$ type, where the magnetic signal should be only at odd $L$-value. It casts serious doubts that such an AFM stacking can explain the origin of the SHG signal[16]. The same conclusion holds for the $++++$ arrangement, not observed in pure $Sr_2IrO_4$, in full agreement with the literature[3,4,9]. To account for the occurrence of an odd-parity hidden order in SHG measurements in pure $Sr_2IrO_4$, we are left with two scenarios. Either the missing AFM orders are light-induced metastable states during the SHG measurements, as suggested in ref. 16, or another kind of magnetic order exists. Let us consider this second scenario now.

**LC order.** The overall experimental procedure to extract the magnetic signal associated with the LC order is detailed in Supplementary Note 2 and it follows methods established in previous works on superconducting cuprates[19,22]. The measured SF intensity, $I_{SF}$ at $\mathbf{Q}=(1,1,2)$, is reported as a function of temperature in Fig. 3a for the pure sample. It is compared with the background baseline $I_{SF}^0$ determined from the measured NSF intensity (see Fig. 3a caption and Supplementary Fig. 3). While $I_{SF}^0$ decreases monotonically as temperature decreases, the SF intensity instead departs from $I_{SF}^0$ below $T_{mag}\approx240\pm30$ K. The difference, $I_{SF}-I_{SF}^0$, evidences a spontaneous magnetic order whose magnetic intensity, $I_{mag}$, is reported in Fig. 3b. It has a different symmetry compared to the conventional AFM phases discussed above and the position corresponds to where magnetic intensity is expected for the LC phase. For the 7% Rh-substituted sample, the same analysis is given in Fig. 3c,d. While the AFM transition is suppressed down to $T_N\approx100$ K, the novel magnetic order is observed at much higher temperature $T_{mag}\approx240\pm30$ K. Within error bars, the transition temperature, $T_{mag}$ and its normalized magnetic intensity do not show a significant difference between the pure system and 7% Rh substitution. This is consistent with the estimate of onset of the hidden order, $T_\Omega$, from SHG (orange squares in Fig. 1a)[15], which does not change appreciably with Rh substitution. Using the calibration of nuclear Bragg peaks intensities (Supplementary Fig. 4), one can deduce the magnetic cross-section of the hidden magnetic order as $\sim2$ mbarns per f.u., which is $<\sim10^{-3}$ of the strongest nuclear Bragg peak. The normalized magnitude of $I_{mag}$ for the hidden magnetic order is similar in both samples and is $\sim5$ times smaller than the one for the AFM order, (as shown by the comparison of the vertical scale of Fig. 3b,d with the red vertical scale of Fig. 2b).

## Discussion

To understand the spontaneous magnetic order at $T_{mag}$, a model is needed to explain the broken symmetries of the hidden order. Concomitant with the SHG data, time-reversal, parity inversion and four-fold rotational symmetries are broken. It preserves the translational symmetry of the underlying lattice, as the magnetic scattering appears on top of the nuclear Bragg peak. The LC model[17,18] is a good candidate, since it can explain all these broken symmetries. Nevertheless, a more detailed and quantitative study is required to establish the exact order parameter in the hidden ordered phase. The weak magnetic cross-section shows how difficult it is to detect and why it was not reported with typical unpolarized neutron diffraction. Due to the experimental limitations, a precise determination of the direction of induced magnetic moments and the full magnetic structure requires further works.

Actually, other magnetic orders, which could potentially account for the SHG data, are not consistent with our finding as they would give rise to distinct scattering patterns. For instance, the proposed structure with $m1'$ point group[15] yields a different relation $H+K+L=2n+1$, as the configuration is opposite when one considers the (1/2, 1/2, 1/2) translation. This breaks the original body-centred nuclear structure[7] and would give rise to magnetic superstructure peaks at $(1, 0, L=2n)$ and no magnetic intensity at any $(1, 1, L)$ position. With Rh doping, one also does not observe magnetic superstructures at $(1, 0, L)$ with even $L$[10,14]. Both points dismiss the proposal of the $m1'$ point group[15].

Alternatively, the SHG measurements have been reinterpreted considering conventional AFM orders[16] with a different stacking along the $c$-axis. We detail in Supplementary Note 3 why the different AFM structures discussed in the literature cannot actually explain our observation. First, since the hidden order occurs at different $\mathbf{Q}$ positions compared to the AFM order, it is clearly distinct with the AF-I order. Second, the AFM state of $2'/m$ symmetry with a $+-+-$ stacking along the $c$-axis of AFM planes is argued to explain SHG data[16]. This phase will give rise to magnetic contributions at positions such as $(1, 0, L=2n+1)$ and nothing at $(1, 1, L)$, where we observe

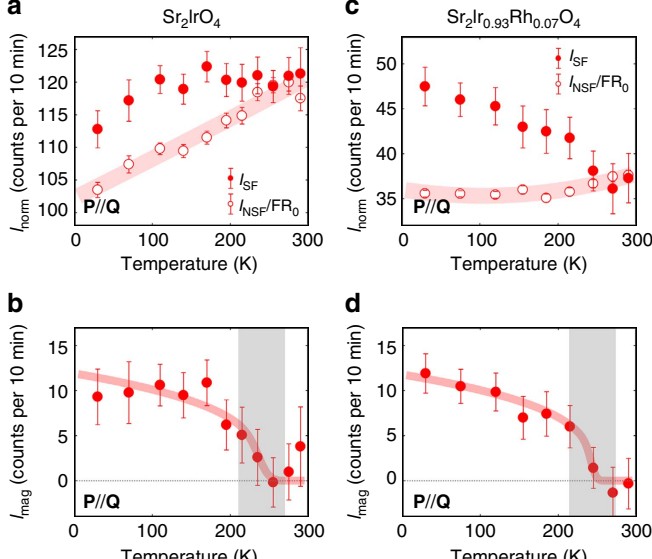

**Figure 3 | Hidden magnetic order in $Sr_2Ir_{1-x}Rh_xO_4$. (a,c)** Temperature dependence of $I_{SF}$ and a bare SF intensity $I_{SF}^0=I_{NSF}/FR_0(T)$ from the polarization leakage. The temperature-dependent bare flipping ratio $FR_0(T)$ is taken from the reference Bragg peak $(2, 2, 0)$ and rescaled to correct the NSF intensity at $(1, 1, 2)$ (see Supplementary Note 2). **(b,d)** Temperature dependence of the magnetic intensity $I_{mag}=I_{SF}-I_{NSF}/FR_0(T)$ at the nuclear Bragg peak $(1, 1, 2)$. Red curves are a guide to the eye and grey areas represent temperature uncertainty in the determination of $T_{mag}$.

the hidden magnetic intensity. Further, the non-magnetoelectric $+ + + +$ phase (AF-II) (described above) could also account for the SHG data[16]. This AF-II phase is reported in the Rh-doped system[10,14] but is absent in the pure system[3,4]. It would also exhibit the largest magnetic contribution at $(1, 0, L = 2n + 1)$ as well as tiny magnetic contributions at Bragg peaks, such as $(1, 1, L = 2 + 4n)$, due to its weak ferromagnetism. This interpretation can be excluded in both samples we have studied. First, in the pure sample, the ferromagnetic order is absent (see Fig. 1)[3,4] and can be only induced by an applied magnetic field of about 0.2 T (2000 Oe)[2]. Second, under Rh substitution, such a ferromagnetic order indeed develops but only below $T_N$ (ref. 10), as it results from the canting of the AFM order, clearly lower than $T_{mag}$. Moreover, the neutron intensity due to ferromagnetism would be $\sim \tan^2\theta \leq 10^{-2}$ smaller than the AFM one, that is, at least one order of magnitude smaller than the observed magnetic scattering we report here. All these arguments allow us to rule out the weak ferromagnetism derived from the canted AFM order as a candidate to account for the observed magnetic scattering at $\mathbf{Q} = (1, 1, 2)$. Therefore, our observation of a magnetic signal at $\mathbf{Q} = (1, 1, 2)$ is not consistent with any kind of stackings along the $c$-axis of the pseudospin AFM orders considered to explain the SHG signal in ref. 16.

Using polarized neutron diffraction, we have experimentally addressed all these alternative phases and found out evidence for a hidden magnetic order in $Sr_2Ir_{1-x}Rh_xO_4$. It is a translation-invariant but time-reversal symmetry broken phase that is consistent with the LC order of $2'/m$ point group symmetry, concomitantly compatible with the SHG signal. In that model, all $IrO_6$ octahedra, which are the building blocks of the material, are identically decorated by the same set of staggered magnetic moments, whose magnetism cancels out on each octahedron (as depicted in Fig. 1b). This magnetic order is then clearly distinct from the AFM one, where each octahedron has a single pseudospin on the Ir site and where the nearest octahedra have staggered moments.

In conclusion, we report the first evidence of an unconventional magnetic order in $Sr_2(Ir,Rh)O_4$, which breaks time-reversal symmetry but preserves translational symmetry of the underlying lattice. By analogy with superconducting cuprates, where a similar kind of order is observed, one can refer to it as an intra-unit-cell order. The new magnetic phase overlaps with parity inversion and rotational symmetry broken phase recently reported using SHG[15]. Both observations can be described by the LC order[17,18] proposed for the pseudogap state in cuprates, where it is well supported by polarized neutron measurements[19,22]. Further, the neutron observation in cuprates is confirmed as well by recent SHG measurements that show a global broken inversion symmetry in $YBa_2Cu_3O_{6+x}$ (ref. 26). This may provide more analogy between the iridates and the high-$T_c$ cuprates, in spite of a different nature of $5d$ and $3d$ orbitals. A noticeable difference here is that the loop order occurs in the insulating compounds at half-filling, whereas in cuprates it is observed in the doped metallic (superconducting) state. Our report generalizes the existence of LC electronic states in oxides.

## Methods

**Coaligned single crystals and magnetization measurements.** We have investigated the pure and 7% Rh-doped $Sr_2(Ir,Rh)O_4$ single crystals grown by flux method at Laboratoire de Physique des Solides (Orsay). Owing to a rectangular cuboid shape of the crystals, several tiny crystals could be coaligned in order to increase the total mass, as shown in Supplementary Fig. 2a. To address the conventional AFM order, the magnetization was measured under a magnetic field $H = 1$ T using Magnetic Property Measurement System. The pure system originally exhibits an AFM order, but it can easily show ferromagnetism, by applying a small external magnetic field $H \approx 0.2$ T ($\simeq 2,000$ Oe)[2]. For instance, the reported magnetic moment deduced from magnetization measurements shown in

Supplementary Fig. 2b corresponds to a net ferromagnetic moment under 1 T. This ferromagnetism originates from a canting of the AFM structure. The transition temperature is taken at the maximum slope in $M(T)$ curve, that is, the minimum of the first derivative of magnetization, d$M(T)$/d$T$ shown in Supplementary Fig. 2b. The pure system shows a sharp transition at $T_N \approx 232$ K and the saturated ferromagnetic moment per Ir ion is $\sim 0.08\mu_B$ at 1 T. On the other hand, the doped one shows a broad transition near $T_N \approx 100$ K and the moment is also reduced to $\sim 0.04\mu_B$. To map out the phase diagram (Fig. 1), the transition temperatures for different doping levels were determined from the previous literatures[12,14], following the same procedure.

**Polarized neutron diffraction.** The polarized neutron diffraction experiments were performed on the triple-axis spectrometer 4F1 located at the Orphée reactor in Saclay (France). The polarized neutron setup was similar to the one used previously for studying cuprates[19–21]. A polarized incident neutron beam with $E_i = 13.7$ meV ($k_i = 2.57$ Å$^{-1}$) is obtained by a polarizing supermirror (bender) and scattered neutrons are measured with a Heusler analyser, which determines as well the final neutron polarization. A pyrolytic graphite filter is put before the bender to remove high harmonics. A small magnetic field of typically 10 G is applied using a Helmholtz-like coil. It is used to change adiabatically the direction of the neutron polarization at the sample position. A Mezei flipper is located before the sample position to flip the neutron spin.

All measurements were done in two different scattering planes, either $H00$-$00L$ or $HH0$-$00L$, where the scattering wave-vector is quoted as $\mathbf{Q} = Ha^\star + Kb^\star + Lc^\star \equiv (H, K, L)$ with $a^\star = b^\star = 1.15$ Å$^{-1}$ and $c^\star = 0.24$ Å$^{-1}$. As emphasized in Supplementary Note 1, in $Sr_2(Ir,Rh)O_4$, the conventional AFM order is expected in the $H00$-$00L$ plane and the LC phase in the $HH0$-$00L$ plane. For each Bragg position, the scattered neutron intensity is measured in both SF and NSF channels that corresponds to two different states of the Mezei flipper, flipper-off and flipper-on, respectively. One defines the flipping ratio FR $= I_{NSF}/I_{SF}$ of the Bragg peaks intensities in both NSF and SF channels. It determines the polarization efficiency of the apparatus. Due to unavoidable neutron polarization leakage from the NSF to the SF channel (imperfect polarization), a value for FR was obtained between 30 and 50 for the pure sample and between 50 and 65 for the doped sample. In order to measure a small magnetic signal on top of the large nuclear peak, it is essential to keep very stable and homogeneous neutron polarization through the whole measurement. Thus all the data were measured at a fixed configuration of the spectrometer while changing the temperature. The neutron measurements were performed with a neutron polarization $\mathbf{P}\|\mathbf{Q}$ where the magnetic signal appears entirely in the SF channel[19,22]. The AF-I order was further studied with a polarization $\mathbf{P}\perp\mathbf{Q}$ (but still within the $H00$-$00L$ plane) as well as with a polarization $\mathbf{P}\|\mathbf{z}$, which is perpendicular to the scattering plane (along $0K0$ in the given case).

**Data availability.** The data that support the findings of this study are available from the corresponding authors upon request.

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

## Acknowledgements

We thank S. Di Matteo, D. Hsieh, B.J. Kim, M.R. Norman and C.M. Varma for fruitful discussions. We also acknowledge financial supports from the projects NirvAna (contract ANR-14-OHRI-0010) and SOCRATE (ANR-15-CE30-0009-01) of the ANR French agency. J.J. was supported by an Incoming CEA fellowship from the CEA-Enhanced Eurotalents program, co-funded by FP7 Marie-Sklodowska-Curie COFUND program (Grant Agreement 600382).

## Author contributions

J.J., Y.S. and P.B. performed the polarized neutron experiments at LLB Saclay. J.J. and P.B. analysed the neutron data. A.L. and V.B. performed the single crystals growth and their caracterization. J.J. co-aligned single crystals. J.J., Y.S. and P.B. wrote the manuscript with contributions from all authors. P.B. supervised the project.

## Additional information

**Competing interests:** The authors declare no competing financial interests.

**Publisher's note**: 

