## [Peer Review File · Nature Communications]

Reviewers' Comments:

Reviewer #1 (Remarks to the Author):

The authors investigate a complex ordering pattern in Sr₂IrO₄ and the Rh-doped version. Using neutron scattering (polarized and unpolarized) they make the case for an intra-unit-cell order including loop current patterns with canceling magnetic signal within each unit cell and conserving translational invariance. This may provide a consistent explanation for the second-harmonic-generation (SHG) signal observed previously in this phase. Together with the characterization of an additional antiferromagnetic phase at lower temperature the authors give a compelling argument for a loop current phase analogous to what has been proposed for the cuprate pseudo gap phase many years ago.

The conclusion is still somewhat speculative and the authors are cautious enough not to overstate their claim. However, I consider this work as a most solid analysis to rule out probably all of the more mundane explanations for the SHG signals. Establishing such a phase would indeed be an important result in the search of exotic states of matter.

The publication is well composed and gives a good introduction to the problem. Also the supplementary material is very helpful to understand certain details. Unfortunately, there are somewhat many typos, misspellings and grammatical errors which should be eliminated.

In conclusion, I think that this manuscript reports exciting results which will have a strong impact on the discussion of perovskite transition metal oxides. In addition it is very timely in view of the increased interest in iridates in various context. Therefore I recommend to publish the manuscript as a nature communication after some minor rewriting.

Reviewer #2 (Remarks to the Author):

The evidence for an odd-parity hidden order or loop current phase in the iridates was first reported in a recent study of Sr₂(Ir,Rh)O₄ using optical second-harmonic generation (SHG) measurements. This novel phase, which is long-sought in cuprates, features the broken time-reversal symmetry and invariant translational symmetry of the underlying lattice. This finding represents an exciting development in studies of spin-orbit assisted iridates. However, there are competing proposals that cast doubt on the new phase, and more importantly, more

investigations are needed to confirm it. This polarized neutron study fills that hiatus. It presents additional and strong evidence that helps eliminate the doubts and confirm the existence of the loop current phase in the iridates. The data are unambiguous and the discussion is sound. This work is therefore an important addition to this field and should be published in Nature Communications.

A few minor comments:

1. Although the overall discussion is clear, there are many abbreviations that are not defined, such as LC, SF, NSF, etc. The authors need to clearly define them both in the text and figures.
2. Some phrases used are not consistent. For instance, "LC" is used in the first half of the paper, but "loop current" is used in the rest.
3. There are some typos. For example, In Fig.1, the Refs for TN'-Magnetization should be 18-20, rather than 14-16; on page 7, last line, it should be "for" rather than "fro", etc.

Reviewer #3 (Remarks to the Author):

There are a number of frustrating grammatical inconsistencies and typographical errors in the manuscript, for example line 5 on page 2 materials should be material, page 7 last line fro should be for, page 8 line 6 an open parentheses. These distract from the science.

The introduction sets the scene for the study. The authors should note that the diffraction studies in references 3-5 use modest resolution diffractometers and cannot be taken as "proof" of a lack of symmetry lowering. Whilst the absence of this is of critical importance for the validity of the subsequent discussion I do not see this as a significant concern.

The experimental studies and their analysis are first class, I suggest the authors clarify the colour coding in Figure 1(c) to identify the Sr cations.

What is unclear to me is the audience. This is written as a physics paper and will be attractive to a physics audience. However it is not immediately accessible to the wider materials science audience. To understand the significance of the paper one needs to be atop of selection rules, understand neutron polarisation and time reversal. I think the authors should revisit the discussion section and attempt to clarify the rationale their experimental approach and explain the significance of these for a non-specialist. Alternatively they should consider a more specialised journal.

Dear Editor,

See Below our detailed replies to the referees's comments point by point.

Reviewer #1 (Remarks to the Author):

The authors investigate a complex ordering pattern in Sr₂IrO₄ and the Rh-doped version. Using neutron scattering (polarized and unpolarized) they make the case for an intra-unit-cell order including loop current patterns with canceling magnetic signal within each unit cell and conserving translational invariance. This may provide a consistent explanation for the second-harmonic-generation (SHG) signal observed previously in this phase. Together with the characterization of an additional antiferromagnetic phase at lower temperature the authors give a compelling argument for a loop current phase analogous to what has been proposed for the cuprate pseudo gap phase many years ago.

The conclusion is still somewhat speculative and the authors are cautious enough not to overstate their claim. However, I consider this work as a most solid analysis to rule out probably all of the more mundane explanations for the SHG signals. Establishing such a phase would indeed be an important result in the search of exotic states of matter.

The publication is well composed and gives a good introduction to the problem. Also the supplementary material is very helpful to understand certain details. Unfortunately, there are somewhat many typos, misspellings and grammatical errors which should be eliminated.

In conclusion, I think that this manuscript reports exciting results which will have a strong impact on the discussion of perovskite transition metal oxides. In addition it is very timely in view of the increased interest in iridates in various context. Therefore I recommend to publish the manuscript as a nature communication after some minor rewriting.

We thank the referee for his/her comments and assessment. We have re-organized the main text and rewrote the supplementary notes. We corrected the typos.

Reviewer #2 (Remarks to the Author):

The evidence for an odd-parity hidden order or loop current phase in the iridates was first reported in a recent study of Sr₂(Ir,Rh)O₄ using optical second-harmonic generation (SHG) measurements. This novel phase, which is long-sought in cuprates, features the broken time-reversal symmetry and invariant translational symmetry of the underlying lattice. This finding represents an exciting development in studies of spin-orbit assisted iridates. However, there are competing proposals that cast doubt on the new phase, and more importantly, more investigations are needed to confirm it. This polarized neutron study fills that hiatus. It presents additional and strong evidence that helps eliminate the doubts and confirm the existence of the loop current phase in the iridates. The data are unambiguous and the discussion is sound. This work is therefore an important addition to this field and should be published in Nature Communications.

A few minor comments:

1. Although the overall discussion is clear, there are many abbreviations that are not defined, such as LC, SF, NSF, etc. The authors need to clearly define them both in the text and figures.
2. Some phrases used are not consistent. For instance, "LC" is used in the first half of the paper, but "loop current" is used in the rest.

3. There are some typos. For example, In Fig.1, the Refs for TN'-Magnetization should be 18-20, rather than 14-16; on page 7, last line, it should be "for" rather than "fro", etc.

We thank the referee for his/her comments and assessment. We have re-organized the main text and rewrote the supplementary notes. We corrected the typos. Some of the abbreviations were done on the first paragraph (in our first submission). It has been rewritten now as an abstract. The abbreviations are now defined in the text and in the figures and some (like IUC) have been eliminated. The references in Fig. 1 have been updated.

Reviewer #3 (Remarks to the Author):

There are a number of frustrating grammatical inconsistencies and typographical errors in the manuscript, for example line 5 on page 2 materials should be material, page 7 last line fro should be for, page 8 line 6 an open parentheses. These distract from the science.

The introduction sets the scene for the study. The authors should note that the diffraction studies in references 3-5 use modest resolution diffractometers and cannot be taken as "proof" of a lack of symmetry lowering. Whilst the absence of this is of critical importance for the validity of the subsequent discussion I do not see this as a significant concern.

The experimental studies and their analysis are first class, I suggest the authors clarify the colour coding in Figure 1(c) to identify the Sr cations.

What is unclear to me is the audience. This is written as a physics paper and will be attractive to a physics audience. However it is not immediately accessible to the wider materials science audience. To understand the significance of the paper one needs to be atop of selection rules, understand neutron polarisation and time reversal. I think the authors should revisit the discussion section and attempt to clarify the rationale their experimental approach and explain the significance of these for a non-specialist. Alternatively they should consider a more specialised journal.

We thank the referee for his/her comments and assessment. We have re-organized the main text and rewrote the supplementary notes. We have labelled in the caption the atoms color of Fig. 1c. In the introduction, we have precise that the spatial resolution of the diffractometers used are too modest to lack of symmetry lowering. This is not related to our work but about the conclusion of ref 15 which clearly motivated us for that experiment.

We have recomposed the discussion section to clarify our conclusion and organized the arguments so that non-specialist can get rapidly the key informations. The supplementary note 5 gives the detailed argumentation for specialist.